# Danish Clinical Named Entity Recognition and Relation Extraction

**Martin Sundahl Laursen**[*]
The Maersk Mc-Kinney Moller Institute
University of Southern Denmark
msla@mmmi.sdu.dk

**Jannik Skyttegaard Pedersen**[*]
The Maersk Mc-Kinney Moller Institute
University of Southern Denmark
jasp@mmmi.sdu.dk

**Rasmus Søgaard Hansen**
Department of Clinical Biochemistry
Odense University Hospital

**Thiusius Rajeeth Savarimuthu**
The Maersk Mc-Kinney Moller Institute
University of Southern Denmark

**Pernille Just Vinholt**
Department of Clinical Biochemistry
Odense University Hospital

## Abstract

Electronic health records contain important information regarding the patients' medical history but much of this information is stored in unstructured narrative text. This paper presents the first Danish clinical named entity recognition and relation extraction dataset for extraction of six types of clinical events, six types of attributes, and three types of relations. The dataset contains 11,607 paragraphs from Danish electronic health records containing 54,631 clinical events, 41,954 attributes, and 14,604 relations. We detail the methodology of developing the annotation scheme, and train a transformer-based architecture on the developed dataset with macro F1 performance of 60.05%, 44.85%, and 70.64% for clinical events, attributes, and relations, respectively.

## 1 Introduction

Electronic health records (EHR) contain important information regarding the patients' medical history including diagnoses, medications, treatment plans, allergies, and test results. However, much of this information is stored in unstructured narrative text. While this information could be used to guide diagnostic decision making and treatment plans, the unstructured format makes it infeasible to fully exploit in clinical practice and research.

Natural language processing (NLP) algorithms could be used to transform the unstructured narrative text of the EHR into structured information and give medical doctors (MD) a fast overview of even a medical history spanning multiple years. NLP models' ability to process and extract information from written text keeps improving with benchmark-breaking models being published on a regular basis. For example, transformer-based models such as GPT-3 (Brown et al., 2020), BERT (Devlin et al., 2019), and ELECTRA (Clark et al., 2020) have recently shown promising results for many NLP tasks, e.g. named entity recognition and relation extraction (NER). In NER, models are trained to tag words with predefined entities and find the relations between them. In clinical NER, entities such as diseases, treatments, drugs, and tests have been extracted automatically from EHRs. However, many of the developed datasets are only in English and for specific clinical specialities or note types (Uzuner et al., 2007, 2010; Bethard et al., 2016).

This paper describes the methodology for developing the first Danish clinical NER dataset. The dataset consists of text paragraphs from Danish EHRs spanning multiple departments and note types.

First, the paper describes the clinical dataset, the strategy for choosing entities tailored to extract important information from EHRs, and the annotation scheme. Next, we train a transformer-based architecture on the developed NER dataset.

## 2 Related works

The annotation schemes and extracted clinical entities and relations vary. Agrawal et al. (2022) extracted medications, their status (active, discontinued, neither), and attributes. The i2b2 2009 challenge (Uzuner et al., 2010) and n2c2 2018 Track 2 (Henry et al., 2020) only extracted medications

---
[*]Equal contribution

and their attributes. Examples of attributes are name, dosage, mode of administration, frequency, duration, reason, strength, form, and adverse drug effects.

SemEval-2016 Task 12 (Bethard et al., 2016) extracted time entities; event entities and their contextual modality, degree, polarity, and type; and temporal relations between time and event entities (before, overlap, before—overlap, after).

SemEval-2015 Task 14 (Elhadad et al., 2015) and CLEF eHealth 2013 Task 1 (Pradhan et al., 2015) extracted disorder mentions and mapped them to their UMLS/SNOMED concept unique identifier. The former also classified attributes such as the disorder's subject, course, body location, and severity, and whether it was negated, uncertain, conditional, or generic.

The i2b2 2010 challenge (Uzuner et al., 2011) extracted entities (medical problems, treatments, tests), assertions (present, absent, possible, conditional, hypothetical future, and associated with someone other than the patient), and relations between medical problem entities and each of medical problem, treatment, and test entities.

The i2b2 2012 challenge (Sun et al., 2013b) extracted clinically relevant events. Their type was classified as concept (problem, test, treatment), clinical department, evidentials indicating source of information, or occurrences (events that happen to the patient). Polarity was classified as positive or negated, and modality as happens, proposed, conditional, or possible. Temporal expressions were extracted with their type (date, time, duration, frequency), value, and modifier indicating whether the temporal expression was exact or not. Temporal relations indicating the type of connections between events and temporal expressions were also extracted.

## 3 Methods

This section describes the data, annotation scheme, and model used for Danish clinical NER.

### 3.1 Data

We extracted 11,607 paragraphs with a length between 11 and 75 words from EHRs from Odense University Hospital in Denmark. Paragraphs were sampled randomly from different EHR note types across every department of the hospital to ensure the data distribution would resemble that of EHRs: 46% were from clinical contacts, 13% primary

| Clinical event | Description |
|---|---|
| **Disease** | A disorder of structure or function, especially one that has a known cause and a distinctive group of symptoms, signs, or anatomical changes. Examples include cancer, influenza, and narcolepsy. |
| **Symptom** | A symptom is a physical or mental feature which is regarded as indicating a condition of disease, particularly such a feature that is apparent to the patient. We include abnormal findings, which the MD makes when examining the patient objectively, as these are sometimes coinciding with symptoms—e.g. bruises. Examples include headache, stomach ache, and pain. |
| **Diagnostic** | Any tool or method concerned with the diagnosis of illnesses or other problems. Includes measurements and tests. Examples include CT scans, blood samples, and temperatures. |
| **Treatment** | A treatment is any medical care given to a patient for an illness or injury. Examples include medication, plaster, and rehabilitation. |
| **Anatomy** | Any part of human anatomy. Includes body fluids and excrements. Examples include arms, organs, and blood. |
| **Result** | All results of diagnostics that do not carry any meaning without being coupled to the diagnostic. Examples include numbers that indicate length, temperature, or volumes. Diseases or symptoms found by diagnostics are annotated as such, e.g. a tumour found by a CT scan. |

Table 1: Description of clinical events. Descriptions were inspired by the Oxford English Dictionary.

journals, 10% care data, 3% epicrises, 3% ambulatory care contacts, 2% surgical notes, 2% emergency room journals, and 20% were from 55 different minor EHR note types. Paragraphs were lowercased and anonymised by two of the authors.

### 3.2 Annotation

#### 3.2.1 Annotation scheme

Two MDs with expert clinical domain knowledge developed the annotation scheme through an iterative process of making annotation rules and testing them.

Annotation rules were made to extract clinically relevant information from the medical history. Focus was for the rules to be as complete as possible to capture all important information about the medical history while still being simple to use for the annotators.

We extracted three types of information: clinical events, the attributes of the clinical events, and relations between the clinical events.

Clinical events were: diseases; symptoms, including abnormal findings; diagnostics; treatments; anatomies including body fluids and excrements; and results. Symptoms and abnormal findings were joined in one as they sometimes coincided. Normal findings were not included as there were so many that they would cloud the visualisation of the history. Table 1 shows all clinical events and their descriptions as defined by the medical experts.

Clinical events were further described by their attributes. Attributes were: prior; current; fu-

| Attributes | Description |
|---|---|
| Prior | Entities that occurred in prior admissions or in the distant past. Includes treatments that are being stopped at that point in time. |
| Current | Entities that occur in the present. Includes prescribed medicine. |
| Future | Entities that occur or might occur in the future—e.g. the risk of skin cancer, or ordering diagnostics for a later day. |
| Doubt | Any entity that is not confirmed. Includes any treatments that might need to be started in the future. |
| Negation | Entities such as diseases or symptoms that are mentioned as not being present. |
| Non-patient | Entities that are not related to the patient in question. One example is the disease history of the patient's relatives. |

Table 2: Description of attributes.

ture; doubt; negation; and non-patient. All clinical events could take one of the six attributes except anatomies and results. Anatomies did not take any attributes while results could only take a prior or current attribute. Table 2 shows all attributes and their descriptions.

Clinical events could connect to each other in limited ways through one-way relations. Diseases, diagnostics, and symptoms could connect to anatomies through a "has location" relation. Diseases, symptoms, and anatomies could connect to treatments through a "is treated with" relation. Diagnostics could connect to results through a "has result" relation.

Figure 1 shows an overview of the clinical events, attributes, and relations. Appendix A shows the full annotation guidelines with further details and explanations to the annotators.

### 3.2.2 Annotation process

Six annotators were recruited for the task. Five were Master of Science in Medicine students and one was a MD.

Figure 2 shows the process of annotator training. It included reading the annotation guide and an iterative process of annotating a learning set of 55 paragraphs (not included in dataset) followed by error analysis until a final test was made on a set of 98 gold paragraphs annotated by an expert MD. Paragraphs were annotated using the CLAMP software (Soysal et al., 2017). We report the micro F1 of each annotator on the gold set.

Figure 3 shows an example of an annotated paragraph.

### 3.3 Entity and relation extraction model

This section describes the architecture of the Princeton University Relation Extraction system (PURE) (Zhong and Chen, 2021) which we used and adapted for Danish clinical NER. It further describes the dataset used and the training of the models.

### 3.3.1 Model architecture

PURE—the 2021 state-of-the-art on entity and relation extraction—is a NER deep learning model based on a transformer structure. The model has a separate entity and relation extraction part.

For entity extraction, the model takes as input all possible text spans up to a maximum length. A transformer extracts contextual word embeddings for the start and end token of each span. They are concatenated with a learned span width embedding and classified by a feedforward network.

When extracting relations, for each candidate pair of entities, the text is passed through a transformer with inserted entity start and end marker tokens for the subject and object entity, also indicating the type. The concatenation of the start marker token for the candidate subject and object entity is classified by a feedforward neural network.

We used PURE's entity extraction approach for clinical events and the relation extraction approach for relations between clinical events.

We used our own approach adapted from the PURE relation extraction approach for attributes. We inserted clinical event start and end marker tokens, passed all tokens through a transformer, concatenated the start and end marker tokens, and classified the attribute using a feedforward network. The marker tokens were used for classification instead of the word(s) forming the clinical event to guide the model to look more at the context rather than the specific word—the context being the important factor in attribute classification. Additionally, enriching the input with the type of the clinical event could guide the model if attributes were described differently for different clinical events.

Figure 4 shows the three types of extraction tasks.

### 3.3.2 Datasets

Table 3 shows the number of clinical events, attributes, and relations by type in the train, validation, and test set. The dataset had a total of 11,607 paragraphs, each containing a varying number of clinical events, attributes, and relations. On average, each paragraph contained 4.7 clinical events, 3.6 attributes, and 1.3 relations. We split the paragraphs in train, validation, and test sets for an approximate 80%–10%–10% ratio between each type of clinical event, attribute, and relation. The sets were unbalanced on type of entity or relation—e.g. for the attributes training

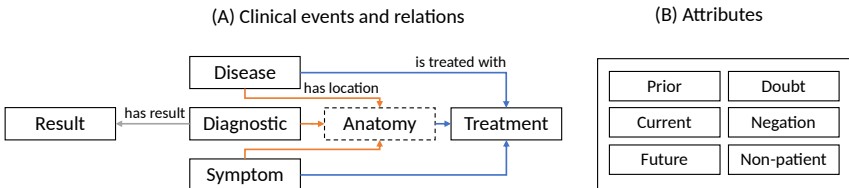

Figure 1: (A) Clinical events and relations between them. Symptoms include abnormal findings. Anatomies include body fluids and excrements. Diagnostics include measurements and tests. Blue: "is treated with". Orange: "has location". Grey: "has result". (B) Attributes. Anatomy (dashed lines) takes no attributes. Other clinical events must take one attribute. Results only take prior or current attributes.

| | Train (% of row total) | Validation (% of row total) | Test (%of row total) | Total (% of column total) |
|---|---|---|---|---|
| **Paragraphs** | 9,687 (83%) | 960 (8%) | 960 (8%) | **11,607 (100%)** |
| | | **Clinical events** | | |
| **Diseases** | 2,033 (78%) | 295 (11%) | 272 (10%) | **2,600 (5%)** |
| **Symptoms** | 11,937 (80%) | 1,455 (10%) | 1,571 (10%) | **14,963 (27%)** |
| **Diagnostics** | 8,921 (80%) | 1,095 (10%) | 1,194 (11%) | **11,210 (21%)** |
| **Treatments** | 6,918 (79%) | 911 (10%) | 882 (10%) | **8,711 (16%)** |
| **Anatomies** | 10,172 (80%) | 1,227 (10%) | 1,278 (10%) | **12,677 (23%)** |
| **Results** | 3,522 (79%) | 473 (11%) | 475 (11%) | **4,470 (8%)** |
| **TOTAL** | **43,503 (80%)** | **5,456 (10%)** | **5,672 (10%)** | 54,631 (100%) |
| | | **Attributes** | | |
| **Prior** | 2,028 (80%) | 237 (9%) | 283 (11%) | **2,548 (6%)** |
| **Current** | 23,217 (79%) | 3,021 (10%) | 3,109 (11%) | **29,347 (70%)** |
| **Future** | 1,237 (79%) | 161 (10%) | 160 (10%) | **1,558 (4%)** |
| **Doubt** | 2,479 (82%) | 263 (9%) | 289 (10%) | **3,031 (7%)** |
| **Negation** | 3,890 (80%) | 496 (10%) | 500 (10%) | **4,886 (12%)** |
| **Non-patient** | 480 (82%) | 51 (9%) | 53 (9%) | **584 (1%)** |
| **TOTAL** | **33,331 (79%)** | **4,229 (10%)** | **4,394 (10%)** | 41,954 (100%) |
| | | **Relations** | | |
| **is treated with** | 1,485 (80%) | 175 (9%) | 197 (11%) | **1,857 (13%)** |
| **has location** | 6,501 (80%) | 779 (10%) | 823 (10%) | **8,103 (55%)** |
| **has result** | 3,652 (79%) | 499 (11%) | 493 (11%) | **4,644 (32%)** |
| **TOTAL** | **11,638 (80%)** | **1,453 (10%)** | **1,513 (10%)** | 14,604 (100%) |

Table 3: Composition of the train, validation and test sets by type of clinical event, attribute, and relation.

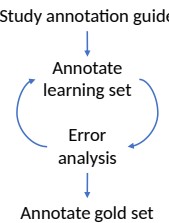

Figure 2: Annotator training process. Figure inspired by Sun et al. (2013a).

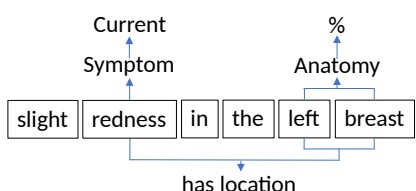

Figure 3: Example of annotated paragraph. % signifies that no attribute could be assigned to the clinical event per the annotation scheme.

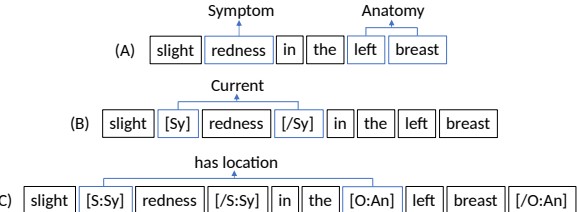

Figure 4: (A) Classification of clinical events from start and end tokens of span. Span width embedding not depicted. (B) Classification of attribute using clinical event marker tokens. (C) Classification of relation using subject/object and clinical event marker tokens. Figure inspired by Zhong and Chen (2021).

| Evaluation metric | Loss | Micro | | | Macro | | |
|---|---|---|---|---|---|---|---|
| | | R% | P% | F1% | R% | P% | F1% |
| Micro F1 | Unweighted | 79.14 | 79.14 | 79.14 | 38.34 | 40.51 | 38.56 |
| | Weighted | 61.81 | 61.81 | 61.81 | 45.35 | 33.20 | 34.23 |
| Macro F1 | Unweighted | 77.30 | 77.30 | 77.30 | 41.88 | 41.90 | 41.48 |
| | Weighted | 60.13 | 60.13 | 60.13 | 51.37 | 41.87 | 43.85 |

Table 4: Validation set micro and macro recall, precision, and F1 score on the attribute extraction task when selecting the best iteration of the model based on micro and macro F1 score with unweighted and weighted loss. 2 hidden layers of size 75 was used for the test. R: Recall. P: Precision.

set, there were 23,217 current and only 480 non-patient attributes. All datasets were in the json format used by PURE (see Zhong and Chen (2021)).

### 3.3.3 Training

When training the clinical event extraction model, we used a Danish Clinical ELECTRA pretrained on the narrative text from 299,718 EHRs from Odense University Hospital as the transformer base (Pedersen et al., 2022). The model had ~13M parameters and consisted of 12 transformer layers with 4 attention heads. We used a dropout of 0.1 after the last ELECTRA hidden layer output. We tested classification heads with two hidden layers of varying size, each followed by a dropout of 0.2 and a ReLU activation function. We used a maximum span of 8 and a train batch size of 32. We trained for 100 epochs using the AdamW optimizer with learning rate 1e-5 for the transformer layers and 1e-4 for the classification head, and a warm-up proportion of 0.1.

When training each of the models for extracting attributes and relations, we used the same transformer base with a normalisation layer and a dropout of 0.1 after the concatenation of tokens. We tested classification heads with two hidden layers of varying size, each followed by a dropout of 0.2 and a ReLU activation function. We further tested a classification head only consisting of a single classification layer. We used a train batch size of 32 and a maximum sequence length of 128. We trained for 20 epochs using the AdamW optimizer with learning rate 2e-5 and a warm-up proportion of 0.1.

We modified the training method of PURE to guide the models towards equal performance on all classes by using a weighted loss function to counteract the unbalanced dataset and chosing the best model for each of the clinical event, attribute,

and relation extraction tasks as the model iteration with the best macro F1 on the validation set, rather than the micro F1 standard of PURE. Table 4 shows a test of the performance on the attribute extraction task when selecting the best iteration of the model based on micro and macro F1 score with unweighted and weighted loss. Using the macro F1 score with weighted loss gave the best performance across all classes. Appendix B shows the confusion matrices for each combination.

Class weights were calculated for the training of each model using the default formula in Scikit-learn (Pedregosa et al., 2011):

$$w_x = \frac{n_{samples}}{n_{classes} \cdot n_x} \qquad (1)$$

where $x$ is the class, $n_{samples}$ is the number of total samples, and $n_{classes}$ is the number of classes. The negative class, i.e. samples not to be given any label by the model, was given a weight of 1.

The negative class was excluded when calculating the F1. We only trained the attribute and relation models to make classifications that were allowed for the connected clinical events according to the annotation scheme. Appendix C shows the results of the hyperparameter search. We report the micro and macro recall, precision, and F1 for the best models on the test set.

## 4 Results

This section presents the agreement of the annotators on the gold set and the results of the Danish clinical NER models.

### 4.1 Annotation

Table 5 shows the annotators' micro F1 performance on the gold set. For clinical events, it ranged 83.71%–91.24% (average 85.62%) for overlapping matches, and 74.12%–85.15% (average 77.67%) for exact matches. For attributes, it ranged 79.21%–86.19% (average 81.71%) and for relations 71.28%–90.06% (average 77.79%).

### 4.2 Entity and relation extraction model

The models that had the best validation performance in the hyperparameter search were:

- A clinical event extraction model with two hidden layers of size 450 in the classification head.

| Annotator | A | B | C | D | E | F |
|---|---|---|---|---|---|---|
| | Overlap match, micro F1% | | | | | |
| Clinical event | 91.24 | 84.22 | 84.41 | 85.71 | 84.43 | 83.71 |
| Attribute | 86.19 | 83.06 | 79.21 | 81.29 | 79.75 | 80.75 |
| Relation | 90.06 | 76.97 | 75.60 | 77.01 | 71.28 | 75.84 |
| | Exact match, micro F1% | | | | | |
| Clinical event | 85.15 | 76.08 | 76.29 | 78.69 | 74.12 | 75.71 |

Table 5: The anonymised annotators' performance on the gold set. Exact match: a match is defined as the exact tokens annotated in the gold set with the same label. Overlap match: a match is defined as minimum one token overlapping with the gold set annotation of the same label. Only an overlap match F1 is calculated for attributes and relations as evaluating an exact match would propagate the potential error in the span of the clinical event to which the attribute or relation is connected.

| | Micro | | | Macro | | |
|---|---|---|---|---|---|---|
| | R% | P% | F1% | R% | P% | F1% |
| | Overlap match | | | | | |
| Clinical events | 66.29 | 77.31 | 71.38 | 64.88 | 72.60 | 68.20 |
| | Exact match | | | | | |
| Clinical events | 60.97 | 65.64 | 63.22 | 59.84 | 61.30 | 60.05 |
| Attributes | 66.04 | 66.04 | 66.04 | 51.60 | 42.64 | 44.85 |
| Relations | 75.88 | 72.66 | 74.23 | 74.74 | 67.85 | 70.64 |

Table 6: Performance of the best clinical event, attribute, and relation extraction models on the test set. Attributes and relations are only reported with an exact match as the models do not consider the span of the clinical event from which the attribute or relation is classified. R: Recall. P: Precision.

- An attribute extraction model with a single classification layer.

- A relation extraction model with two hidden layers of size 150 in the classification head.

Table 6 shows the performance of the best models on the test set. Clinical events were extracted with exact micro F1 63.22% and macro F1 60.05%, attributes with micro F1 66.04% and macro F1 44.85%, and relations with micro F1 74.23% and macro F1 70.64%. The negative class was excluded when calculating the recall, precision, and F1 scores.

Figure 5 shows the confusion matrices of performance on clinical events, attributes, and relations. The confusion matrices include the clinical events and relations that were not extracted and falsely extracted by the model ('O').

The model for clinical event extraction performed best on anatomies (69%) and worst on results (53%). 1,568 spans were falsely extracted

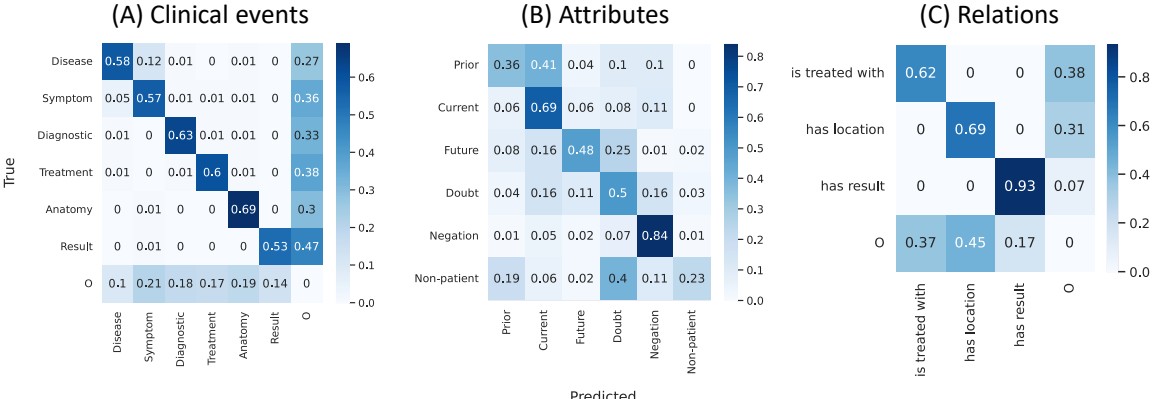

Figure 5: Confusion matrices of performance on (A) clinical events, (B) attributes, and (C) relations. 'O' counts the clinical events and relations that were not extracted and falsely extracted by the model.

as a clinical event with symptoms being the most frequent (21%). The model for attribute extraction performed best on negations (84%) and worst on non-patient (23%). The model for relation extraction performed best on "has result" (93%) and worst on "is treated with" (62%). 432 false relations were extracted of which "has location" was the most frequent misclassification (45%).

## 5   Discussion and limitations

This paper presented a methodology for developing a dataset for Danish clinical NER. It presented an annotation scheme for annotation of all clinical events, their attributes, and relations that are relevant for the medical history. The dataset included text paragraphs from Danish EHRs spanning multiple departments and note types.

We trained and adapted PURE NER deep learning models to extract clinical events (overlap match macro F1 68.20%; exact match macro F1 60.05%), attributes of clinical events (macro F1 44.85%), and relations between clinical events (macro F1 70.64%). The results are promising for Danish clinical NER but need improvement. A discussion of possible improvements to the methodology, limitations, and future work is provided below.

The clinical event extraction model had similar performance on all classes with accuracies between 53% (results) and 69% (anatomies). There was little contamination between classes as most errors were caused by failure to extract or false extraction of a clinical event. There was some contamination between symptoms and diseases with 12% of diseases being classified as symptoms and

5% of symptoms being classified as diseases. This supports claims by annotators that diseases and symptoms in some cases are difficult to differentiate and that extra attention must be given to differentiate these in the annotation guidelines.

The attribute extraction model had large differences in performance with accuracies between 23% (non-patient) and 84% (negation). There were more misclassifications of the non-patient attribute as doubt (40%) than correct classifications. The future and doubt attributes had significant contamination between them with 25% and 11% misclassifications as the other class, respectively. The many misclassifications between non-patient and doubt attributes, and especially future and doubt attributes, could indicate that the model would improve if the non-patient, doubt, and future attributes were merged to a single class of uncertain attributes. This would most likely not harm the usefulness of the model to MDs significantly.

The fact that more prior attributes were misclassified as current (41%) than correct classifications (36%) likewise indicates that these two attributes could be merged into a single class of clinical events that occurred. This would, however, decrease the usefulness of the model as it is important for MDs reviewing the medical history to know if a clinical event is prior or current.

The relation model extracted 93% of the "has result" relations, and 62% and 69% of the "is treated with" and "has location" relations, respectively. The differences are likely caused by the fact that the "has result" relation only connects diagnostics to results while the two other relations have three different one-way relationships.

In this paper, we only explored one type of NER model and tested a limited set of architectures and hyperparameters. Future work could include testing other architectures and enriching the model input with more information, e.g. the output of a text parser, which could help differentiate attributes dealing with the time-aspect. The six annotators had an average micro F1 (overlap match) of 85.62%, 81.71%, and 77.79% for clinical events, attributes, and relations, respectively. Merging certain attributes and more emphasis on differences between symptoms and diseases could increase these scores.

The Danish clinical NER dataset is not made publicly available due to it containing sensitive information. We advise interested researchers to contact us for sharing possibilities.

## 6  Conclusions

This paper presented methodology and annotation scheme for developing the first Danish clinical NER dataset. The corpus consists of 11,607 paragraphs annotated for six entity types, six attributes, and three relations. The corpus was used to fine-tune language models which showed promising results for classifying the entities, attributes, and relations of the dataset.

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

# Appendices

# A  Annotation guidelines

## A.1  Clinical events

### A.1.1  Disease

Contains all diseases including diseases that could be considered a result of a Diagnostic.

### A.1.2  Symptom

Includes all symptoms and abnormal findings. Findings that are not abnormal should not be annotated. However, a negation of an abnormal finding should be annotated because the abnormal finding is mentioned even though it is not present. For example, "fracture" should be annotated in the sentence "there is no sign of fracture."

If there is a negation of a non-abnormal finding, it should be annotated in the entity. For example, "cannot hear" is annotated in the sentence "patient cannot hear anything."

In the sentence "no symptoms," the word "symptoms" should not be annotated as a symptom, as it does not contain any information.

In case a symptom or abnormal finding is found by a Diagnostic, there may be a coincidence with the Result entity. Here, it is annotated as Symptom if the entity can provide sufficient meaning alone. For example, "cyst" or "tumour."

If the Symptom cannot stand alone and one needs to know which Diagnostic was carried out in order to understand the result, the entity should instead be annotated as Result and have a "has result" relationship from the Diagnostic entity. For example, this applies to "Temp: 24 C" and "Stix: 3+". "Temp" and "Stix" are annotated as Diagnostic with "is treated with" relationship to Result "24 C" and "3+."

### A.1.3  Result

Includes all results of Diagnostic, e.g. values and blood test results.

A Result cannot stand on its own. A relation from the Diagnostic is needed for it to make sense. These can be entities like "stable", "positive", "negative", "24 C" or "3+".

Typically, this entity will appear in sentence structures with a colon: "Diagnostic: Result". Note that the two entities are mentioned very close to each other in the text—in this case only with a colon in between. An example could be "Temp: 24 C" or "Stix: 3+". "Temp" and "Stix" are annotated as Diagnostics with a "has result" relation to Result "24 C" and "3+".

Entities that can instead be annotated as Symptom will typically be mentioned further away or completely lack a Diagnostic as a Symptom can stand alone and make sense.

See also the description for Symptom.

### A.1.4  Diagnostic

Includes all diagnostics, measurements, and tests. This can include CT scans, blood tests, MR scans, and recordings of a newborn's length, temperature, etc.

Note that "blood sample results" and "radiology description" are not a Diagnostic and should not be annotated.

If KAD is mentioned along with a volume, e.g. "KAD emptied of 200 mL," it is marked as Diagnostic - Result. If there is no volume specified, KAD is annotated as Treatment.

### A.1.5  Treatment

Includes all forms of treatment including medication.

To annotate entities as concisely as possible, for example in the sentence "good effect of 2.5 mg morphine IV," only "morphine" should be annotated as Treatment.

In the sentence "treated for xxx," the word "treatment" should not be annotated as Treatment as it does not contain any information.

If KAD is mentioned without a volume indication, it should be annotated as Treatment. If KAD is mentioned with a volume, for example "KAD emptied for 200 mL," it should be annotated as "Diagnostic - Result."

### A.1.6 Anatomy

Includes all mentions of anatomies and things from the body (blood, feces, urine, sweat, etc.).

Typically used to indicate the location of a Disease or Symptom, a Diagnostic, or a Treatment. Examples: "brain", "left foot" or "duodenum".

When Anatomy is described by an adjacent word, for example "left", this should be included in the entity.

Remember to annotate the Anatomy entities that should not be linked to other entities.

## A.2 Attributes

### A.2.1 Current

The entity is either present, carried out, or current. If medication is prescribed to the patient, this should also be marked as "Treatment - Current", as it can be assumed that the treatment will start and it may be the last time it is mentioned in the journal. On the other hand, "Scheduling a CT for Tuesday." should be marked as "Future" as it will be described in a future medical note, for example with the result.

### A.2.2 Negation

The entity is not present. For example, if it is mentioned that the patient does not have a fracture, the fracture should be marked as Symptom - Negation. Note that the word "not" should not be part of the marked entity. However, if there is a negation of a normal finding, it should be annotated as such. For example, "cannot hear" in the sentence "patient cannot hear anything" is annotated as Symptom - Present.

### A.2.3 Prior

If the entity refers to a previous case, i.e., a previous hospitalisation or if it happened a long time ago. For example, it should be annotated as a prior Treatment when a cast or drain is removed, as the treatment is finished. However, if a CT scan from the previous day is mentioned, it should be annotated as Current.

### A.2.4 Future

Everything that takes place in the future. For example, cancer is annotated as Disease - Future if it is mentioned that "there is a risk of cancer if you use tanning beds too often."

It is marked as Diagnostic - Future if an MRI scan is planned for the next day. However, if it is written "the treatment with xxx starts" or "rp. xxx" it should be marked as Treatment - Current as it is assumed that the treatment will certainly happen.

Also includes references to possible future treatments.

### A.2.5 Doubt

If the patient might have a disease that has not yet been confirmed.

If a Treatment should be given provided that certain things change.

The difference between Doubt and Future is that Future is more certain - it is going to happen - while Doubt is more uncertain or conditional.

### A.2.6 Non-patient

If an entity does not have a direct connection to the patient. This can occur when a general letter is sent out regarding cancer screening. Cancer should then be annotated as Disease - Non-patient. If it is mentioned that the patient's mother had a certain disease, it should also be annotated in this way.

## A.3 Relations

When entities are annotated, the relationships between entities can be annotated. This is done by pulling the "From entity" over to the "To entity". The direction of the relationship is important. Therefore, pay attention to the name of the relationship and read it out loud if necessary, "Entity - Relation - Entity" and listen to see if it makes sense or if the arrow needs to be reversed. CLAMP will show which relationships can be annotated for the pair being drawn between.

**has location**
    From entities: Disease, Symptom, Diagnostic.
    To entities: Anatomy.

**has result**
    From entities: Diagnostic.
    To entities: Result.

**is treated with**

From entities: Disease, Symptom, Anatomy.

To entities: Treatment.

The "is treated with" relation links the entities Disease, Symptom, and Anatomy to a Treatment. In some cases, sentences describing a required treatment could be linked to both an Anatomy and Treatment entity. In this case, the Treatment should be linked to the Symptom instead of the Anatomy. You should only link the Anatomy to the Treatment using the "is treated with" relation if the Treatment cannot be linked to anything else. Example: "Left knee skin scraping is treated with plaster." Annotation: skin scraping - "Treated with" - plaster.

### A.4 General notes

It is important not to annotate periods, commas, etc. unless they are part of an abbreviation. For example, in "Patient has cancer," only "cancer" and not "cancer." should be marked. If you double-click a word, CLAMP will only mark the word and not any punctuation next to the word. This can make it a bit troublesome to include periods in abbreviations.

Entities should be annotated as concisely as possible without losing meaning. This means that in the sentence "there are signs of cancer," only "cancer" and not "signs of cancer" should be marked as an entity. If an entity has some describing words next to it, the following rule can be used to decide how much should be annotated. In the sentence "pain in the front of the arm," only "arm" is marked as Anatomy since "front" and "arm" are connected through the word "of." In the sentence "pain in the left arm," "left arm" is marked as Anatomy since there are no words between "left" and "arm". In sentences describing a prescription of medication, only the name is marked as Treatment, and not, for example, the quantity indication or the number of days.

Entities may not overlap with each other.

### B Selection of loss and evaluation metric

Figure 6 shows the confusion matrices for the attribute extraction task when selecting the best iteration of the model based on micro and macro F1 score with unweighted and weighted loss.

Using the micro F1 to select the best iteration of the model resulted in some classes being prac-

| | Classification head hidden layers | Validation Exact F1 % |
|---|---|---|
| **Clinical event** | 2x 75 | 58.49 |
| | 2x 150 | 59.82 |
| | 2x 300 | 60.68 |
| | 2x 450 | 61.34 |
| | 2x 600 | 60.91 |
| **Attribute** | None | 48.01 |
| | 2x 50 | 43.20 |
| | 2x 75 | 43.85 |
| | 2x 150 | 44.10 |
| | 2x 300 | 44.32 |
| **Relation** | None | 66.15 |
| | 2x 75 | 68.39 |
| | 2x 150 | 68.85 |
| | 2x 300 | 67.39 |

Table 7: Results of the hyperparameter search.

tically excluded during classification. Using the macro F1 to select the best model iteration and training with a weighted loss gave the most equal performance on all classes.

### C Hyperparameter search

Table 7 shows the results of the hyperparameter search.

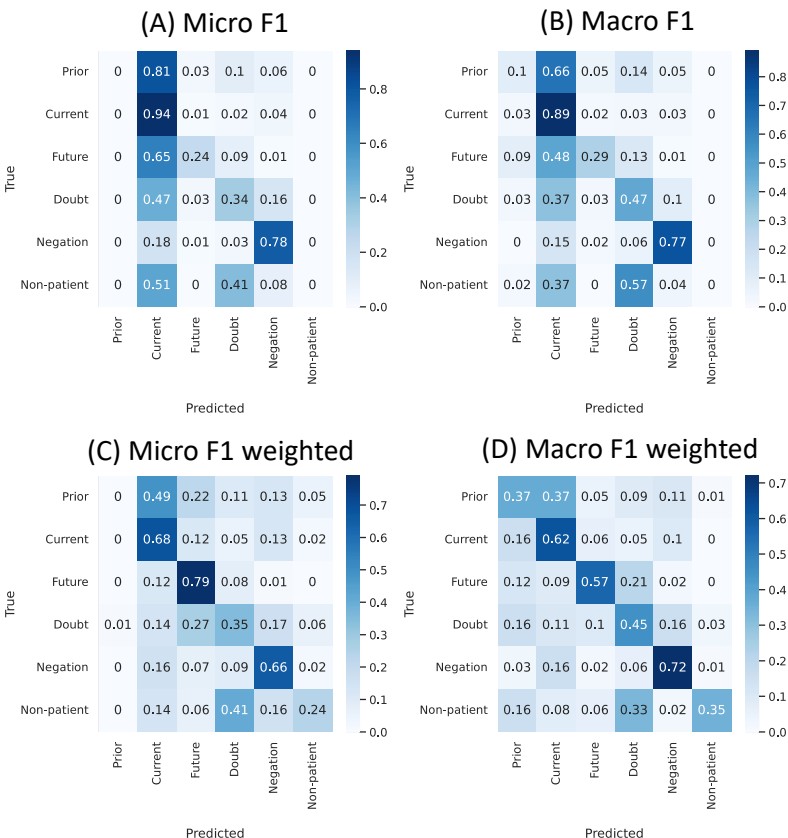

Figure 6: Confusion matrices showing the attribute extraction validation performance of the models chosen based on (A) micro F1, (B) macro F1, (C) micro F1 trained with weighted loss, and (D) macro F1 trained with weighted loss.