# OpenReview forum: "Danish Clinical Named Entity Recognition and Relation Extraction"
_NoDaLiDa/2023/Conference — NoDaLiDa 2023_

### Official Review · Reviewer_vS91 · 2023-03-05
**A new Danish NER dataset of electronic health records annotated for clinical events, attributes, and relations.**

**Rating:** 8
**Confidence:** 4

**Review:**

Pros:
1. A useful and clearly explained methodology, easy to follow for anyone who might attempt the same in another language
2. A dataset for an under-represented language such as Danish
3. Great annotation process with iterative steps and a "training" phase that must have led to error minimisation (and with a good explanation for cases of contamination which is understandable for this field)
4. Very detailed future work that will certainly benefit the field.

Cons:
1. Due to GDPR this dataset cannot be shared, which is understandable. Perhaps an attempt on pseudonymization or sanitization might allow for part of it to be released to aid the scientific community?
2. The annotation guidelines were not released but I was wondering if the fact that MDs drafted them and medically related people annotated them introduced any kind of bias to the dataset
3. A lot of information in the Appendix was actually quite interesting and useful. The text could at places have been shorter to allow for that. (e.g. lengthy description of training in 2.3.3 could have been a Table of parameters which is usual)

All in all, great paper I can see much use of it.

**Paper Type:**

Long paper

---

### Official Review · Reviewer_f57d · 2023-03-08
**Well-written paper introducing a novel dataset for entity and relation extraction within the clinical domain for EHR written in Danish.**

**Rating:** 9
**Confidence:** 4

**Review:**

The paper makes a strong contribution to its field by introducing a novel dataset with entities and relations extracted. It is well-written and a joy to read. Solid and well-documented experimental setup. Thorough discussion and evaluation.

*Pros*
- Well-written and clear
- Strong contribution
- High quality content and evaluation methods

*Cons*
- None that I can identify

*Comments*
- Would it make sense to put the hyperparameters of the traning process(es) in a table, instead of in running text?
- Negation is an important attribute, but also difficult to identify. Why do you exclude it from your results throughout the paper?
- Is there any difference in performance on different EHRs types?

**Paper Type:**

Long paper

---

### Official Review · Reviewer_BiVY · 2023-03-11
**A novel dataset and NER model for the Danish medical language**

**Rating:** 8
**Confidence:** 4

**Review:**

Authors have created and present the first Danish clinical named entity recognition and relation extraction dataset of a considerable size. They have developed and tested a concise and clear annotation scheme and methodology that other research groups can adapt to create similar datasets for other languages.

Much effort has been invested to acquire a representative dataset (incl. representative splitting into train/dev/test sets) and to annotate it consistently. It is not clear, however, why the authors have selected only 11.6k paragraphs from all the available EHRs (should be much more available in the hospital's archive?).

Regarding the annotation scheme, there are no references to related work - why the scheme and methodology was specified from scratch instead of adopting some scheme and methodology used e.g. for English or other languages? Some comparison is required.

Based on the created dataset, authors have trained and evaluated a specialised NER and relation extraction model by adapting the state-of-the-art PURE framework for entity and relation extraction. Authors have not at all discussed and justified their choice and any possible alternatives. Otherwise, the model training process and settings are well described.

The annotators' performance on a gold set is quite low, given the very limited ontology (annotation scheme) - 77.7-85.6%. This is reflected in the model's performance. Some comparison with (references to) related work on other languages / similar datasets is required.

Good discussion on limitations and possible improvements in Section 4.

Well written and inspiring paper!

Minor:
- Incorrect opening quotation marks everywhere in the paper.
- Description of the experiment in Appendix B overlaps quite a lot with Section 2.3.3; add some missing bits from B to 2.3.3, and B (incl. Figure 6) can be removed.
- Appendix C is also not very informative and could be removed due to page limits.

**Paper Type:**

Long paper

---

### Decision · Program_Chairs · 2023-03-17

Accept